# Laser Direct Writing of Dual-Scale 3D Structures for Cell Repelling at High Cellular Density

**DOI:** 10.3390/ijms23063247

**Published:** 2022-03-17

**Authors:** Irina Alexandra Paun, Bogdan Stefanita Calin, Roxana Cristina Popescu, Eugenia Tanasa, Antoniu Moldovan

**Affiliations:** 1Center for Advanced Laser Technologies (CETAL), National Institute for Laser, Plasma and Radiation Physics, RO-077125 Măgurele, Romania; bogdan.calin@inflpr.ro (B.S.C.); eugenia.tanasa@physics.pub.ro (E.T.); 2Faculty of Applied Sciences, University Politehnica of Bucharest, RO-060042 Bucharest, Romania; 3Horia Hulubei National Institute for Physics and Nuclear Engineering IFIN-HH, RO-077125 Măgurele, Romania; roxana.popescu@nipne.ro; 4National Institute for Laser, Plasma and Radiation Physics, RO-077125 Măgurele, Romania; antoniu.moldovan@inflpr.ro

**Keywords:** laser direct writing, two-photon polymerization, dual-scale structure, cell repellent

## Abstract

The fabrication of complex, reproducible, and accurate micro-and nanostructured interfaces that impede the interaction between material’s surface and different cell types represents an important objective in the development of medical devices. This can be achieved by topographical means such as dual-scale structures, mainly represented by microstructures with surface nanopatterning. Fabrication via laser irradiation of materials seems promising. However, laser-assisted fabrication of dual-scale structures, i.e., ripples relies on stochastic processes deriving from laser–matter interaction, limiting the control over the structures’ topography. In this paper, we report on laser fabrication of cell-repellent dual-scale 3D structures with fully reproducible and high spatial accuracy topographies. Structures were designed as micrometric “mushrooms” decorated with fingerprint-like nanometric features with heights and periodicities close to those of the calamistrum, i.e., 200–300 nm. They were fabricated by Laser Direct Writing via Two-Photon Polymerization of IP-Dip photoresist. Design and laser writing parameters were optimized for conferring cell-repellent properties to the structures, even for high cellular densities in the culture medium. The structures were most efficient in repelling the cells when the fingerprint-like features had periodicities and heights of ≅200 nm, fairly close to the repellent surfaces of the calamistrum. Laser power was the most important parameter for the optimization protocol.

## 1. Introduction

Micro- and nanoscale surface engineering can induce major improvements to material performances, by changing the materials’ properties, such as morphology, chemical composition, structure, frictional forces, adhesive properties and wettability [1]. In biomedical applications such as implantable devices for controlled drug delivery [2,3] and scaffolds for tissue engineering [4,5,6], the surface properties are extremely important, as they come into direct contact with biological fluids, cells and tissues from the implantable area. 

To date, it has been demonstrated that a cell’s attachment to a surface is influenced not only by the chemistry of the surface but also by its topography [7]. In this regard, a major concern is to obtain reproducible, complex and accurate micro- and nanostructured interfaces able to control the cell’s attachment. In particular, limiting and reducing the interaction between the material’s surface and different types of cells is very important for the development of new devices such as biosensors [8], blood-interacting devices [9], anti-microbial surfaces [10], and so on, where the adhesion of microorganisms such as cells or bacteria would restraint the functionality of these devices. Thus the employed materials require a selected or null interaction with plasma proteins and with living cells, in order to assure the maximum functionality of the final medical device. The most straightforward manner that this can be achieved is by changing the topography of the cells (or other microorganisms)–material interfaces [11].

Special attention is currently being dedicated to dual-scale structures having spatial features of different size scales, i.e., microstructures with nanopatterns on their surface. Over the years, several two-step fabrication methods for dual-scale microstructures have been reported, such as molding and film deposition [12,13], additive manufacturing, electrospinning [14], nanoparticle coating on microstructured surfaced [15] and others.

One of the most popular technologies for modifying the topography of a material is laser irradiation. Of particular interest for the present study, laser surface micro- and nanopatterning evolved into a standalone domain, with extensive applications [16,17,18,19,20,21,22,23]. The laser-generated micro- and nanostructures have been used for a wide variety of applications such as to modify the surface wettability [24] and to control cellular adhesion [25]. More importantly, laser irradiation of materials provides the tools for the single-step fabrication of dual-scale structures [26,27,28,29]. To date, laser-assisted fabrication of dual-scale structures relied on stochastic processes deriving from laser–matter interactions [27]. This further limited the control over the geometric characteristics (size, shape, orientation of nanopatterns), while also showing an increased sensitivity towards the processing parameters (incident wavelength, laser power, laser pulse overlap, writing speeds). 

Recently, an innovative laser processing technology has been developed, which can provide results that are impossible to achieve via simple laser irradiation of materials. This technology is known as Laser Direct Writing via Two-Photon Polymerization (LDW via TPP), or 3D lithography. It is an additive manufacturing method that provides 2D and 3D features with complex architectures, sub-micrometer resolution, having lateral spatial features as low as 90 nm. In this paper, we report laser-based dual-scale 3D microstructures with fully controllable geometry, obtained by LDW via TPP. Recently, the scientific community has migrated towards research and applications that take advantage of the high resolution of the method, such as the development of micrometric and submicrometric biomimetic features [30,31,32].

The design of biomimetic features is inspired by certain elements that occur in nature. Laser engineering provided a way to obtain revolutionary micronic, submicronic and nanometric structures, with spatial resolution up to 90 nm, that accurately reproduce micro- and nanofeatures encountered in nature [1]. Such biomimetic structures, entirely or partially, exhibited specific, i.e., desired properties, such as non/low cellular adhesion [33,34,35]. For example, the calamistrum of the cribellate spider, Chinese pangolin scales with longitudinal ridges [35], the scales on a shark’s skin [36] and other elements of living organisms served as inspiration for non-adhesive nano- and microstructured surfaces.

In the last years, LDW via TPP has been successfully used for obtaining 3D architectures, in both micrometric and submicrometric ranges, capable of controlling the attachment of cells [37,38,39]. Recently, we have reported hierarchically mushroom-like constructs decorated with micrometric and submicrometric ripple-like features, that were able to impede the cellular attachment, when working at standard (i.e., relatively low) cellular densities in the culture medium, i.e., 5000 cells/sample [37]. That study addressed a broader application/concept, that was to obtain biomimetic dual-scale 3D structures with anti-adhesive properties, inspired by the size and geometry of the calamistrum of cribellate spiders [33], with a deviation in periodicity and height of less than 30%.

In the present study, we used Laser Direct Writing Via Two-Photon Polymerization (LDW via TPP) of IP-Dip photoresist for the fabrication of dual-scale 3D structures in the shape of mushrooms “decorated” with fingerprint-like submicrometric features. The structures will be named Nanostructured Mushroom-like Pillars, NMP. The geometric characteristics were designed to join micro- and nanostructuring into 3D constructs with fully reproducible and controllable architectures (micrometric “mushrooms” decorated with fingerprint-like nanofeatures having specific periodicity). The obtained structures differed from all other laser-induced dual-scale fabrication methods reported so far that relied on stochastic laser–matter interaction [26,27,28] or were confined to specific geometric shapes [29] or both.

We particularly aim to optimize the structures’ design and the laser writing parameters (laser power, scan speed) for improving the cell-repellent properties of the structures at high cellular densities in the culture medium (i.e., cellular densities in the culture medium up to five times higher than we previously used). The present study opens up new perspectives regarding the potential of the cell-repellent structures, such as for the development of biosensors that require poor or no cell adherence to the sensor surface [40], of implantable chips where adherent proteins and cells might cause the inflammation of the implantation site [41,42,43], as well as for the development of antibacterial surfaces [44,45].

## 2. Results and Discussion

### 2.1. Structures Fabrication and Characterization

#### 2.1.1. Structures Fabrication by Laser Direct Writing via Two-Photon Polymerization

The NMP structures were fabricated by laser direct writing via two-photon polymerization LDW via TPP using a Nanoscribe 3D Lithography system. Details about the experimental method can be found in Section 4, Section 4.1. LDW via TPP is a type of 3D printing technology that allows the fabrication of complex 3D structures with practically no limitations regarding the achievable architectures and having submicrometric spatial accuracy, i.e., down to 90 nm and full reproducibility of the design. 

The laser power was determined through extrapolation from a calibration curve, which is determined before each laser processing step. The voltage on the acousto-optic modulator is scanned throughout the whole working interval, while the output laser power is measured using a photodiode, before the objective lens.

#### 2.1.2. Structures 2D Morphological Characterization by Optical Microscopy

We first imprinted several areas of 200 × 200 μm^2^ each, where the NMPs were arranged in hexagonal lattices. In order to verify the overall stability and reproducibility of the structures, we kept constant the design and the writing parameters (laser power, scanning speed), as shown in Figure 1a. We first checked the reproducibility and stability of the 3D structures (Figure 1a). It can be observed that most of the structures preserved their vertical position and spatial arrangement in the form of hexagonal lattice on the underlying glass substrate. The only exception, marked with a red star, was observed for the lower part of Figure 1a. In this case, the NMPs slipped on the glass substrate and lost their spatial arrangement in the hexagonal lattice as initially set by the design. We also checked the stability and reproducibility of the structures using the same design and different writing parameters (laser power, scanning speed), as shown in Figure 1b. Figure 1b shows optical images of NMPs arrays fabricated by LDW via TPP using various laser powers (upper panel) and scanning speeds (lower panel). With increasing scanning speed and decreasing laser power, the structures were less stable on the glass substrate. This most probably happened because of a low degree of polymerization. The most obvious effects in this regard are marked with red stars in Figure 1b. With increasing laser speed and decreasing laser power, the structures were less stable on the glass substrate, as expected, due to a lower degree of polymerization. From left to right, it can be observed that an increasing number of NMPs lost their original position on the glass substrate, most probably because the laser energy dose delivered to the photoresist was too low. A lower energy dose means a lower polymerization degree, which further means increased flexibility of the polymeric structures, during and after sample development. A low polymerization degree means that traces of unpolymerized material are lodged between polymeric chains of irradiated material. These traces can either escape the irradiated material during sample development, inducing bending and shrinking, or simply lower the stiffness of the microstructures after sample development and drying. This phenomenon destabilized the structures during the developing and drying processes. From these preliminary qualitative observations, we concluded that in the interval 80–120 μm/s the scanning speed has a negligible influence on the overall stability and reproducibility of the dual-scale structures. On the other side, we observed that the laser power is a much more relevant parameter. Consequently, in the further optimization experiments, the scanning speed was kept constant at 100 μm/s and only the laser power and the structures’ design were changed.

#### 2.1.3. Structures 2D Morphological Characterization by Scanning Electron Microscopy

In order to visualize in more detail the morphology of the structures as a whole and their positioning on the glass substrate, the NMP structures were characterized by scanning electron microscopy (SEM). A similar parametric study as in the optical images from Figure 1 was performed, for checking in more detail the role of the writing parameters on the morphology of the structures. We mostly focused on the fingerprint-like nanostructures that decorate the “mushroom’s hats”, because they are expected to have the highest influence over cellular attachment via a non-adhesive behavior [33].

Figure 2 illustrates scanning electron micrographs (SEM) of NMPs fabricated by LDW via TPP with different laser powers. From upper to lower panels, the SEM images were recorded with increasing magnifications that allowed us to observe the samples’ morphology in more detail. It can be seen that, for laser powers between 11 and 12 mW, the NMPs were not well attached to the glass substrate and some of them were slightly dislocated from their original position. In addition, for this irradiation regime, some NMPs lost their vertical position and bent slightly, as observed in the upper panel of Figure 2. As far as the top views of individual NMP from Figure 2 middle panel, the “mushroom hats” were nearly the same for all NMPs, regardless of the laser power, having a circular shape and diameter of 15 μm. A better view is shown in Figure 2 lower panel, where it appears that, for laser powers between 11 and 12 mW, the “mushroom’s hats” showed less surface undulation and the fingerprint-like features were less evidenced. These effects could be assigned to an insufficient polymerization of the photopolymer, which resulted in NMPs structures being less resistant to the developing and drying processes. It could also be the case that inside the “mushrooms” the photopolymerizable material remained in liquid form, i.e., non-polymerized. All these could have caused the bending of NMPs and the less evidenced fingerprint-like features on top of the mushrooms. By increasing the laser power to 13.25 mW, the structures as a whole were much more stable on the glass substrate and the fingerprint-like feature decorating the top of the mushrooms became more visible. Increasing the laser power above 13.25 mW led to local explosions of the photopolymer and consequently to structure destruction (data not shown here).

We report laser powers with two decimals accuracy, as for example 13.25 mW. Such accuracy is atypical for any power measurement device. The powers we report in this study were extrapolated from the calibration curve determined before laser processing, which explains the possibility to achieve two decimal accuracy for the laser power.

In the lower panel of Figure 2, the nanopatterning of the “mushroom’s hats” can be observed, proving the dual-scale nature of the structures. The nanopatterns had geometric characteristics that were precisely designed in a fingerprint-like shape, with specific periodicity, totally different from all the other laser-induced dual-scale fabrication methods reported so far that relied on stochastic laser–matter interaction [26,27,28] or which were confined to specific geometric shapes [29] or both.

Figure 3 shows the influence of samples’ design on the fingerprint-like features. The figure shows scanning electron micrographs (SEM) of nanostructured mushroom-like pillars (NMPs), fabricated by LDW via TPP of IP-Dip photopolymer, where the writing parameters were kept constant and the design/geometry was changed. Specifically, the initial spiral step (noted with h) was multiplied with sub- and supra-unitary coefficients, for the *z*-axis, as indicated above each image. The general overview of the samples shown in the upper panel of Figure 3 did not evidence a straight dependence of the NMPs on the scaling of the spiral step. However, the closer views from the middle and lower panels of Figure 3, going from right to left, indicate that with increasing multiplication coefficient, the fingerprint-like features become more evident. For sub-unitary multiplication coefficients (the first two images going from right to left), the “mushrooms’ hats” were smooth. With increasing supra-unitary multiplication coefficients (the last three figures going from right to left), the fingerprint-like features were more evident.

#### 2.1.4. Structures 3D Morphological Characterization by Atomic Force Microscopy

Atomic force microscopy (AFM) investigations yielded further information about the size range size of the fingerprint-like features on the mushrooms’ hats. For all NMPs, in agreement with the qualitative information delivered by SEM, the AFM images indicate that, regardless of the laser writing parameters, the top of the mushroom-like pillars exhibited quasi-periodic nanostructures similar to fingerprints (Figure 4). Towards the center of the mushrooms’ hats, the periodicity of the fingerprint-like features was smaller and increased towards the edges of the mushrooms. The quantitative evaluation of the height, width and aspect ratio of these features is a very important aspect to be considered. Estimation of the reproducibility of the laser-imprinted structures is equally important. To achieve these, we performed AFM measurements on individual NMPs as a function of the laser power (Figure 4 upper panel). The results confirm the qualitative observations derived from SEM analysis, indicating that the laser power strongly influences the heights and the periodicity of the fingerprint-like features. With increasing laser power from 11 to 13.25 mW (going from left to right in Figure 4), the fingerprint-like features became more evident. For quantitative evaluation of the fingerprint’s height and periodicity, Figure 4 lower panels show 2D plots along the line profiles from the 3D AFM images (red segments Figure 4 upper panel). Given that the fingerprint-like structure is quasi-periodical, for a systematic comparative analysis of the writing parameters, we measured the largest depth and largest height from each 2D profile of each NMP. As expected, the height and the periodicity of the fingerprint-like features depended on the laser irradiation dose delivered to the photopolymer. With decreasing laser power, the fingerprint-like features became finer, most notably due to the smaller size of the volume pixel (voxel, i.e., the irradiated volume of photoresist that undergoes the chain polymerization process). For laser powers between 11 and 11.5 mW, the fingerprint-like features showed small undulations, with no preferential orientation. Their height and periodicity varied randomly between 10–60 nm and 300–800 nm, respectively, in accordance with the disordered topography observed for these samples. With further increase of the laser power to 13.25 mW, the fingerprint-like features become a little more regular, the periodicity decreased from about 700 nm to about 220 nm, and their height ranged between 30 and 140 nm, but with no particular trends. It is important to note that, for NMPs fabricated with 13.25 mW laser power and a scan speed of 100 μm/s (Figure 4 left lower side, marked with a green oval), the height of the fingerprint-like features fell into the range encountered in the spiders’ calamistrum, i.e., 220 nm in height. Figure 4 lower panel displays the heights of the nanostructures as a function of the incident laser power. The heights of the nanostructures show a tendency to increase with the laser power, and also the aspect ratio. However, for the highest laser power, the height is very low. This is a result of the increasing volume of the voxel while maintaining the same distance between neighboring voxels, which significantly decrease the height of the fingerprint-like nanostructures.

From the AFM analysis, we can observe that nanopatterning of dual-scale structures fabricated by LDW via TPP, while controllable with regards to both size and shape, lost its accuracy when reaching very low spatial features over large areas when compared to other laser-based methods [20]. This is determined for the most part by the type of material involved, as the polymeric material used in LDW via TPP undergoes slight changes during the development and drying processes. 

A typical standard deviation for AFM measurements such as those on the NMPs largely depends on the investigated region, as we observed and stated that the height and periodicity of the fingerprint-like features vary randomly. This implies that the statistical deviation varies between small or large ripples, between different areas of the mushroom’s hat where the measurements were performed, and so on. All these aspects have been earlier addressed when describing the SEM and AFM investigations. In all cases, we can state that the standard deviation on fingerprint-like features’ heights is typically between 5 and 10 nm and the standard deviation for their periodicity is from about 50 to 100 nm.

The reason for which the heights and the periodicities of the fingerprint-like features randomly varied is that, although at a micrometric scale the LDW via TPP technique enables very accurate and reproducible writing of 3D structures, when going down to nanoscale structures imprinted on large areas the resulting features lose their spatial accuracy. The reason beyond this effect mainly relies on the fact that the polymeric material used in LDW via TPP undergoes slight changes during the development and drying processes (i.e., shrinking, swelling and/or bending), that cannot be fully controlled even for controlled post-processing experimental conditions, i.e., sample developing, as described in the Section 4 [20].

Figure 5 shows AFM images of NMPs fabricated using the same writing parameters (13.25 mW laser power and 100 µm/s scanning speed) and with different geometries/designs (different spiral steps, i.e., we modified the rate at which the spiral radius changed). The role of this study was to get more information (thus a better control) on the degree of voxel overlapping, with the final goal of controlling the height and periodicity of the fingerprint-like features by means of an interplay between the laser power, scanning speed and design, i.e., overlapping voxels. 

The upper panel in Figure 5 shows AFM images (enhanced colors) of NMPs fabricated by LDW via TPP of IP-Dip photopolymer using the same writing parameters (13.25 mW, 100 μm/s) and different designs/geometries. The role of the multiplication factor on the spiral step (h) on the *z*-axis is clearly evident in the upper panel. With increasing multiplication factors, the fingerprint-like features were more pronounced (going from left to right, upper panel of Figure 5). The middle panel of Figure 5 shows 2D profiles along radial segments across the structures imaged in the upper panel (the red segments). The best results were obtained in terms of fingerprint-like features’ height (200–210 nm), which were marked with green ovals in the three plots on the right of the middle panel. The lower panel illustrates the designs/simulation of the laser path with different scaling of the spiral step in the *z*-direction (side view of the pillar lid). Figure 5 lower panel presents the heights of the nanostructures as a function of the spiral step. The spiral step determines a better spacing between neighboring pixels, which in turn avoids unwanted effects such as polymer bending, welding and remanent electrostatic forces after sample development and drying. Due to these effects, it can be observed that beyond a spacing of 0.45 μm the structures were more “stable” in what concerns their height. While the height was stabilized, the distance between neighboring spirals was increasing with increasing spiral height step, which further means a lower aspect ratio of the resulting nanostructures.
Figure 4(**Upper panel**) Atomic force microscopy images (AFM) (enhanced colors) of dual-scale structures in the shape of nanostructured mushroom-like pillars (NMPs) fabricated by LDW via TPP of IP-Dip photopolymer, having the same design and fabricated using different laser powers; (**Middle panel**) 2D profiles along the radial segments marked in red in the upper panel. For all samples, the scanning speed was 100 μm/s; (**Lower panel**) Nanostructures’ heights as a function of incident laser power step (the error bars are the statistical deviations). The greeen oval from the right part of the middle panel points out the best result in terms of laser power optimization.
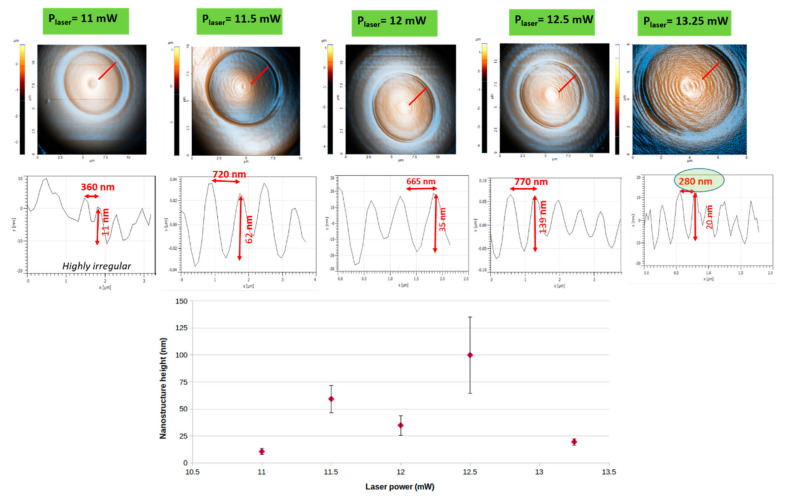


As general observations, we mention that the 2D line profiles extracted from the AFM data and plotted in Figure 4 and Figure 5, were leveled in order to exclude the influence of the parabolic curvature of the mushroom’s hats. The height and quasi-periodicities of the fingerprint-like features do not arise from self-organization or phase transformation, as in the case of “traditional” ripples produced by laser ablation. Instead, in LDW via TPP, the periodicity and height of the fingerprint-like features are determined by: the writing parameters, which in turn determine the size of the voxel (specific for a given microscope objective used for focusing the laser beam into the non-polymerized material, that in our case was a 100× objective), by the design of the structure that settles the overlapping of the voxels and also by the post-processing, i.e., developing procedure, that could affect the overall voxel size and overlapping.

In our study, the spiral step, h, determines both the periodicity of the fingerprint-like features, as well as the voxel overlap and structural integrity of the whole NMP structure. The design algorithm made the spiral follow the exact shape of the surface that it described. For technical reasons, the geometry was designed in Cartesian coordinates. As such, after each 360-degree turn, the spiral had a fixed height difference, Z, which means that the rate of change of the radius was calculated so that the height difference has a fixed value. Any change of the height difference, Z, modified the rate of change of the spiral radius, which in turn changed the periodicity of the fingerprint-like features. Identical writing parameters were used for all samples (laser power 13.25 mW, scanning speed 100 µm/s). In this case, the best results were obtained in terms of height and aspect ratio, which were obtained for laser power 13.25 mW, scanning speed 100 µm/s and spiral step scaling in the z-direction multiplied by 3.
Figure 5(**Upper panel**) Atomic force microscopy images (AFM) (enhanced colors) of dual-scale structures in the shape of nanostructured mushroom-like pillars (NMPs) fabricated by LDW via TPP of IP-Dip photopolymer using the same writing parameters and different designs/geometries; (**Middle panel**) 2D profiles along radial segments across the structures imaged in the upper panel; (**Middle panel**) Simulation of the laser path with different scaling of the spiral step in the z-direction (side view of the pillar lid). For all structures, the laser power was 13.25 mW and the scanning speed was 100 μm/s. (**Lower panel**) Nanostructures’ heights as a function of spiral step (the error bars are the statistical deviations).
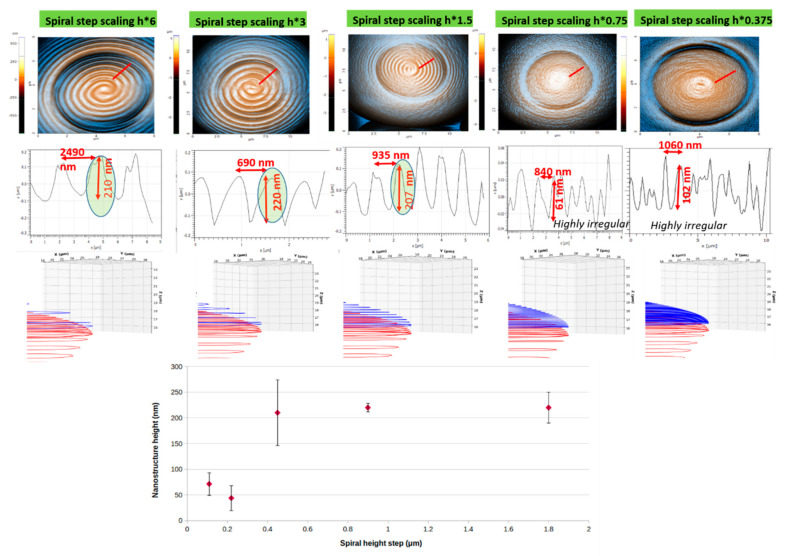


SEM and AFM analysis indicate that the top of each mushroom has two parts: an inner quasi-circular area having quasi-periodical fingerprint-like features, and an outer “ring” where the fingerprints were either missing completely or they were highly irregular, with a random spatial distribution. These observations stand for all NMPs structures, regardless of the writing parameters and design. Several hypotheses about their possible origin can be advanced. For example, the fact that the NMPs were fabricated using a stationary laser beam and a moving sample, using a continuous spiral, means that structures were moved in a circular pattern throughout the writing process. This moving pattern may determine oscillations of the pillars that are top-heavy (i.e., the mushroom’s hats) and have a slim support (i.e., the mushrooms’ legs). The stages movement and the pillar oscillation could have entered in a resonance process for a short time, which could explain the reproducibility and asymmetry of the indentation. 

The scientific motivation of the present study directly derives from the concept of biomimetics. It has been shown that the fingerprint-like nanopillars on the cribellate spiders are able to prevent the adhesion of the nanofibers that the insect produces to make its net [33,46]. This phenomenon is explained by a reduction of the van der Waals forces (adhesive forces) between the two components, leading to a low stickiness process. In reality, the architecture of the spider’s calamistrum morphology is a ripple-like structure with a determined periodicity and height [33].

Eukaryote cells use fiber-like extensions of the cytoskeleton called phyllopodia to form adherence interactions with the extracellular matrix. In a similar manner, it is thus reasonable to expect that a quasi-periodical structure such as our fingerprints, with periodicities and heights as close as possible to that of the calamistrum, would induce a cell-repellent behavior. In our approach, the role of dual-scaling means that because the nanometric fingerprints were fabricated on top of micrometric mushroom-like pillars, both have the role of decreasing the adhesion points for the seeded cells.

At a first glance of the samples through SEM analysis, it seems that for laser powers between 11 and 12 mW, the laser-imprinted 3D structures looked nearly the same. However, such a first clue, as provided by SEM analysis, is not sufficient for concluding on the 3D samples and in particular on the fingerprint-like features from the top of the mushrooms. For more detailed and quantitative observations, we further analyzed our samples by AFM (Figure 4and Figure 5). To correctly understand the information provided by each of these techniques (namely SEM and AFM), we must take into account the different aspects revealed by each of these two techniques. SEM only provides 2D information about the samples’ morphology, while for sub-micrometric going to nanometric scales, SEM images do not always provide enough depth of field for bringing into the light every detail of the samples; in particular, if those details vary in the vertical plane (as in case of the fingerprint-like features from the mushrooms’ hats). This lack of accuracy in SEM images might be induced by non-uniform gold coating of the samples prior to SEM investigation (note that, when the material to be analyzed is non-conductive, as it is the case of the IP-DIP photopolymer, gold coating of the samples is always needed prior to SEM, as we already described in Section 4). For samples with complex 3D architectures as the NMPs, the coating with gold might not be uniformly applied all over the sample’s surface and therefore the SEM images might miss some of the samples’ details. This was the reason we also performed AFM investigations on the samples (Figure 4 and Figure 5). AFM investigations showed that the NMPs and especially the fingerprint-like structures from the top of the mushrooms had a clear dependence on the laser power (please see Figure 4 lower panel, where the height and periodicities of the fingerprint-like features were measured from the 2D plot profiles). Briefly, the main argument for showing both SEM and AFM images for NMPs fabricated at the same laser processing parameters for both investigations was that it is obviously easier for the reader to complete the information given by SEM with the information provided by AFM if the samples are fabricated using the same experimental parameters. Within our study, SEM and AFM are somehow complementary concerning the samples’ structural integrity. That is, SEM provides a more overall image of the samples (i.e., SEM images can be recorded at both small and high magnifications, as can be seen in Figure 2 upper panel versus Figure 2 lower panel), whereas AFM can only be used to investigate very small areas (up to 15 × 15 µm^2^) and to mainly observe 3D nanometric features for the samples surfaces (as observed Figure 4 upper and lower panels, respectively).

#### 2.1.5. Surface Geometry of NMP Structure: 2D and 3D Characterizations and Theoretical Fitting

In order to analyze the surface geometry on an NMP structure, as determined by the voxel shape and overlap, we selected a series of points situated on a line that crosses the sample through the center. The radial surface plot can be observed in Figure 6 that shows: scanning electron micrograph (a) and atomic force microscopy (b) images of a nanostructured mushroom-like structure (NMP) and (c) the cross-section of the indentation surface that goes through the center of an NMP (star-like points: experimental data; continuous lines: parabolic fit). In LDW via TPP, the voxel has a prolate shape with a 2:1 height–width aspect ratio for the optical setup we used for fabrication (microscope objective 100×, having N.A. = 1.3 and immersed in IP-Dip photoresist). Therefore, a cross-section that goes through the long axis of a voxel is determined by an ellipse. In order to extract information from the surface profile, we fitted the experimental data with the upper part of an ellipse, using the least-squares method:(1)$$z=c_{z}+\sqrt{r_{z}^{2}\left(1-\frac{{\left(x-c_{x}\right)}^{2}}{r_{x}^{2}}\right)}$$
where z and x are the Cartesian coordinates for height and radius, c_x_ and c_z_ are the coordinates of the ellipse center, and r_x_, r_z_ are the radii.

Good structural integrity of the NMPs is obtained if neighboring lines have a strong overlap. The rate of change of the spiral radius, however, is not constant. The surface is parabolic, and the height step is constant, therefore determining a variable rate of change of the spiral radius. This was done to accommodate the parabolic envelope function that defines the surface. The strong overlap, however, affects the ability to extract information from fitting and extrapolation, because only a small part of the voxel surface can be analyzed. AFM measurements do not offer enough data points to fit with a parabolic function and extrapolate the approximate voxel shape. Either the fitting parameters do not converge or extrapolated data offer erroneous information. Even so, voxels from the outer edge of the indentation indicate an aspect ratio close to 1:1 height–width, while voxels closer to the center maintain the general 2:1 height–width aspect ratio. This suggests that the voxel is stretched at the point where the indentation is formed.

Although the measurements of the heights and widths of the fingerprint-like features were performed in the inner part of the structures, we also showed the outer part because it, together with the inner part, will be in contact with the cells seeded for in vitro tests. In addition, although not yet measured in their outer parts, the illustration of the whole fingerprint-like structures will be used as a starting point for further in vitro studios that will test the repellent potential of the structures developed in the present study against other cell types and bacterial cultures.
Figure 6(**a**) Scanning electron micrograph (SEM) of a nanostructured mushroom-like pillar (NMP) inclined at 30 degrees; (**b**) Atomic force microscopy (AFMJ) image of NMP from (**a**); (**c**) Cross-section of the indentation surface through the center of a mushroom-like pillar (star-like points: experimental data; continuous lines: parabolic fit).
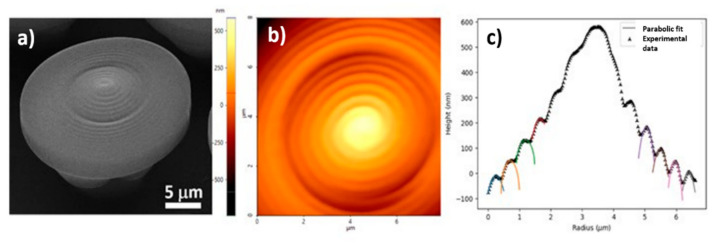


The design of the structures combines the superhydrophobic characteristics of top-heavy microstructures [47], through the mushroom-like pillars, and controllable surface nanopatterning [48], through the spiral-like nanopatterning of the surface.

The LDW using TPP technique has been used before in the 3D structuring of biomimetic surfaces, such as intelligent and transforming microactuators inspired by natural plants [49], mosquito-like microneedles [50,51]. In the present study, we employed the LDW via TPP technique in order to fabricate complex dual-scale 3D structures inspired by the structure of the calamistrum of the cribellate spiders, which are well known for their cell-repellent potential [33].

### 2.2. In Vitro Studies

To study the optimized dual-scale 3D structures (NMPs) fabricated by LDW via TP regarding their in vitro cell-repellent potential, we used primary rat glial OLN 93 cells as an in vitro culture model of oligodendrocytes. The cells were grown in an incubator under standard conditions of temperature and humidity (37 °C, 5% CO_2_). Notably, the NMP structures proved efficient anti-adhesive properties in an extremely humid environment, such as the one provided by the incubator, which is suitable for cell cultures, as compared to other cribellate spider calamistrum inspired structures [52], where the non-adhesive behavior was dependent on the environmental humidity. 

In our previous work reported in [37], the in vitro tests were carried out using a relatively low cell density, i.e., 5000 cells/sample, which led to about a 75% reduction in the adherence of the cells to the NMP structures, as compared to flat glass surfaces. In the present study, we used a cell density five times higher, i.e., 25,000 cells/sample and we checked if the optimized NMP structures preserved their cell-repellent potential. 

The optimized NMP structures will be further named NMPLP for optimum laser power and NMPD for optimum design, according to the experimental findings from Section 2.1. We recall that the scanning speed showed no significant influence over the morphology of the NMPs, thus for the samples prepared for the in vitro experiments, it was kept constant to 100 µm/s. The NMP_LP_, as well as the NMP=D structures, were tested in vitro regarding their ability to reduce the number of adhering cells as compared to flat surfaces. Figure 7 shows comparative experimental results regarding the shape, the area and the number of cells attached on NMPs with optimum design (NMP_D_) as compared to the samples fabricated using optimum laser power (NMP_LP_).

Both parameters, i.e., structures’ design and laser power contributed to the decrease of the cells’ adhesion as compared to the flat surfaces. Equally important, this effect was observed even for high cellular densities as those used in the study. The best results related to our goal, i.e., cell-repellent structures, were obtained for NMPs with optimized design (NMPD), which reduced the cells adhesion by about 60% as compared to flat surfaces (Figure 7i). The cell shape (Figure 7h) and the area covered by cells (Figure 7g) were also influenced by the design and the laser power, but the differences between the effects induced by NMPLP as compared to NMPD were less pronounced. 

We then looked into more detail about cells morphology. In the neural tissue of living organisms, the oligodendrocytes are able to form connections with the neurons, through filopodia–axons interactions in the myelin sheaths [53,54]. Thus, the cytoplasm extensions by means of filopodia are a particular feature of healthy oligodendrocytes. In our experimental study, the cells on the glass surface adhered randomly and presented a normal spindle-like morphology and extended filopodia (Figure 7a). On the other side, the cells from NMPLP and NMPD structures adhered to a much less extent and presented a morphology with a tendency to get close to a circle, with very few, if any, filopodia (Figure 7b,c). In the latter case, the cells did not have a certain orientation and they did not penetrate among the structures, but they stood above them. On both NMPLP and NMPD structures, large gaps were observed in the cellular network and the cells were mostly rounded, with no filopodia extensions, which is a clear sign of weak cell adhesion to the substrate [55]. 

To find some possible answers for our experimental data, we checked the scientific literature concerning the relationship between the morphology of oligodendrocytes and the environmental conditions. We found a strong correlation between these two, in the sense that the morphology of the cells is severely influenced by the environmental conditions, which act as a key factor in cellular differentiation [54,56]. The architecture of the cell culture substrate has been previously shown to influence cells differentiation [57,58,59]. Here, the elongated and flattened polygonal shape of the OLN 93 cells grown on the glass substrate (Figure 1a), is associated with the differentiation of cells towards the mature form in which they fulfill their physiological function of producing the myelin sheath [56]. Conversely, the round shape observed on NMPLP and NMPD structures (Figure 7b,c) is associated with a cell-repellent substrate that prevents cell adhesion [55]. We analyzed the cells’ shape by the Image J software as “circularity”. A circularity value of 1.0 indicates a perfect circle; as the value approaches 0.0, it indicates an increasingly elongated polygon (Figure 7h). 

Due to 3D displacements of the final resulting sample, i.e., cells disposed at different depths on 3D (tall) mushroom-like structures, the quality of the fluorescence images from Figure 7a–c is quite poor. This happens because the fluorescence microscope did not allow a sharp simultaneous focus on different objects situated at different depths (in our case, the cells on the complex 3D mushroom-like pillars). Despite this limiting factor, the fluorescence images from Figure 7a–c were very helpful to obtain information about the cells’ number density and shape. More specifically, from the fluorescence images from Figure 7a–c, we were able to extract these data from a very simple image processing algorithm, using ImageJ software (described in the Section 4, Section 4.2.6 Image Acquisition and Processing). The results of this image processing algorithm, which exclusively evidences the cells, are displayed in Figure 7d–f.
Figure 7Fluorescent images of cells on: (**a**) glass (control), (**b**) nanostructured mushroom-like pillars with optimized laser power (NMPLP), (**c**) nanostructured mushroom-like pillars with optimized design (NMPD); green: structures’ autofluorescence; blue: cells nuclei, Hoechst; red: cells cytoskeleton, Phalloidin; (**d**–**f**) image processing by Image J showing only the cells’ outlines; (**g**–**i**) cells area, cells circularity and reduction of adhering cells for flat, NMP_LP_ and NMP_D_ structures, respectively. The scale (40 µm) from (**f**) is valid for all figures, i.e., (**a**–**f**). The sign “*” from (**g**–**i**) indicate that the data are statistically significant.
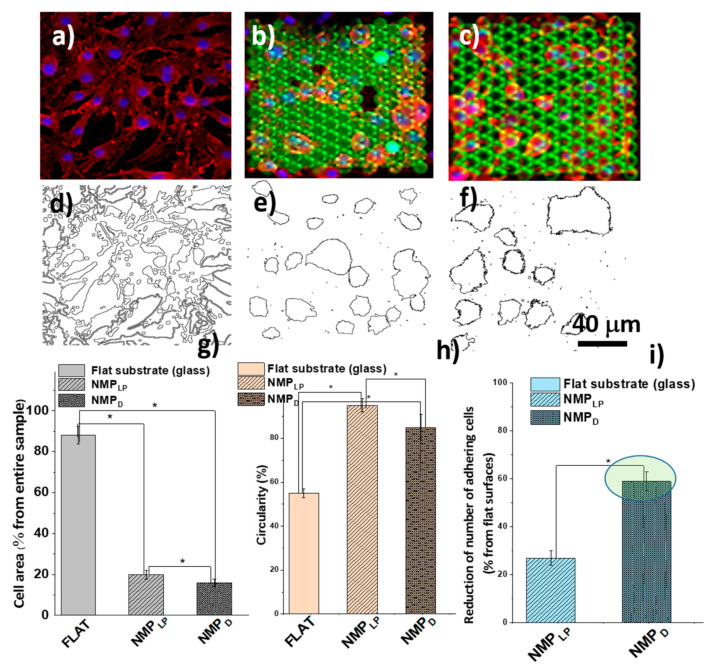


Dual-scale structures for repelling cells have been previously developed. Muck M et al. [60] obtained micro/nanostructured surfaces by the femtosecond laser writing in ASTM F136 Ti-6Al-4V, followed by anodization treatment. This resulted in the obtaining of a modified cell-repellent surface with hundred-nanometer morphological elements. The same technique was used to obtain micrometer-sized patterned surfaces by Heitz et al. [41] for the development of pacemakers. The femtosecond laser writing technique was employed to “write” Au surfaces with antibacterial-adhesion properties [44], which resulted in 1D-rod-like structures of less than 6 µm width, formed by 10 nm elements. The 3D dual-scale structures of polyaniline-gold micro nanopillars were obtained and coated with an antibacterial copolymer (4-vinylbenzyl chloride and 2-(dimethylamino)ethyl methacrylate) [61]. In this case, the best anti-adhesive properties were exhibited by 500 nm–1 µm distanced pillars with heights below 100 nm.

In summary, from the analysis of NMP dual-scale 3D structures having optimized design and laser power and from the evaluation of the cell coverage on the NMP structures, it could be seen that, even in the case of the high cellular density, these optimized NMPs retained their cell-repellent properties.

## 3. Conclusions

We were able to fabricate, by Laser Direct Writing via Two-Photon Polymerization (LDW via TPP) of IP-Dip photoresist, cell-repellent dual-scale 3D structures made of micrometric mushroom-like pillars decorated with nanometric fingerprint-like features, with reasonably controllable heights and periodicities. We investigated the role exerted by the laser power and of the 3D design on LDW via TPP fabrication of the dual-scale structures and the implications for the cellular behavior on the laser-imprinted structures. We found that when the fingerprint-like features had periodicities and heights close to 200 nm, the dual-scale structures were the most efficient in repelling the seeded cells. The structures’ design was a more important parameter for the cell-repellent property of the dual-scale structures, as compared to the laser writing parameters, i.e., laser power, scanning speed. Equally important, the cells repellent property of the dual-scale structures was preserved even for high densities of the seeded cells.

In future work, we foresee establishing a more precise correlation between the 3D design and the writing parameters in terms of overlapping voxels in the LDW via TPP technique, for obtaining better control over the periodicity and height of the dual-scale structures.

## 4. Materials and Methods

### 4.1. Structure Fabrication

Structures were fabricated using Laser Direct Writing via Two-Photon Polymerization (LDW via TPP) [62,63,64]. The main underlying physical phenomenon is two-photon absorption, which primarily targets the ionization of photoinitiator molecules found in a specialized photoresist. The photoresist is a viscous liquid that is drop-casted onto a flat glass substrate. It is generally comprised of photoinitiator, monomer, solvent and other substances that define its physical parameters, such as viscosity and refractive index. The photopolymer solution is irradiated with femtosecond laser pulses. The photoresist is highly transparent for the central wavelength of the incident laser radiation while presenting high absorption for the second harmonic. The transverse intensity profile is Gaussian and the incident laser is strongly focused on the sample using high magnification microscope objectives. Two-photon absorption processes strongly depend on the volumetric intensity of the incident radiation. Therefore, the average laser power is chosen so that the laser pulses propagate through the photopolymer with virtually no interaction, up until the focal point. In the focal point, the laser intensity density surpasses the threshold after which two-photon absorption processes become dominant. This happens in a small, defined volume, called voxel (short for “volume pixel”). As mentioned previously, two-photon absorption targets the ionization of photoinitiator molecules, which become free radicals that initiate a chain reaction polymerization process. After irradiation, the sample is immersed in an appropriate solvent to wash the non-irradiated material, leaving behind the microstructure. The laser processing system was aquired from Nanoscribe,(Eggenstein-Leopoldshafen, Germany). It comprised an Er-doped fiber laser, terminated with a periodically poled LiNbO_3_ crystal, that delivered 120 fs laser pulses, with a repetition rate of 80 MHz, centered on a wavelength of 780 nm. The beam delivery system was terminated with an inverted Zeiss microscope. The microscope objective used in laser processing is a Zeiss Plan-Neofluar 100× NA 1.3. For the microstructures presented in this work, we used a 100× microscope objective with a numerical aperture of 1.3. In order to maintain the laser spot size and position of the focus, the microscope objective was immersed in the liquid photoresist during laser processing. This technique not only provides a constant refractive index throughout the laser propagation but also increases the numerical aperture, therefore obtaining a smaller laser spot in the focus. The photopolymer we used was IP-Dip (Nanoscribe, Eggenstein-Leopoldshafen, Germany), which is a liquid photoresist specially formulated for LDW via TPP using an immersed microscope objective.

The laser direct writing via two-photon polymerization LDW via TPP of the NMP was carried out using the Nanoscribe 3D Lithography system. The material used for structure fabrication was IP-Dip, a specially designed photoresist for Nanoscribe’s novel Dip-in Laser Lithography (DiLL) technology. IP-Dip serves as immersion medium and photosensitive material at the same time, by dipping the microscope objective into this liquid photoresist. Due to its refractive index matched to the focusing optics, IP-Dip guarantees the best focusing and the highest resolution for DiLL technology. The structures were designed as inverted cone-like shapes with parabolic walls (the side walls had an inwards curvature, while the top surface had an outwards curvature). The top diameter was 15 μm, the base diameter was 7 μm and the total height was 18 μm. However, the resulting structures presented slight size variations due to the size of the voxel, which in turn was determined by the laser power (variations of ±1 um for the diameter, ±2 um for the height). Polymer shrinkage and bending following the development process could further affect the overall shape and size of the structures. The fabrication of the structures by LDW via TPP was conducted using laser writing speeds of 80–120 um/s and laser powers of 11–13.25 mW.

### 4.2. Structure Characterization

#### 4.2.1. SEM

Morphological information about the 3D structures was obtained using scanning electron microscopy (SEM) using a Quanta Inspect F50 system (FEI, Eindhoven, The Netherlands), with a field emission gun (FEG) with 1.2 nm resolution. Prior to SEM analysis, the structures were coated with a thin gold film of several nanometers thickness.

#### 4.2.2. AFM

Micro- and nanoscale morphology measurements of the structures were carried out with a commercial atomic force microscopy system (AFM, XE100—Park Systems, Suwon, Korea). The measurements were performed in non-contact mode, using standard cantilevers (AC240TS—Olympus Corporation, Tokyo, Japan; NCHR—Nanosensors, Neuchatel, Switzerland).

For a systematic and comparative analysis, we investigated the fingerprints starting from the center of each mushroom to the first circle, i.e., surface indentation. Given the limited vertical travel of the AFM tip, the scans in the AFM images were restricted to areas of a maximum of 12 × 12 μm^2^. Specifically, on a case-by-case basis, areas of 10 × 10 μm^2^ areas or 8 × 8 μm^2^ areas were scanned. These areas were sufficient for representative and accurate measurements of the fingerprint-like features.

#### 4.2.3. Cell Culture

We used OLN 93 cells as an in vitro culture model of oligodendrocytes. The cells were grown in an incubator under standard conditions of temperature and humidity (37 °C, 5% CO_2_) in DMEM supplemented with 10% fetal bovine serum.

#### 4.2.4. Sample Preparation and Cells Seeding

The samples were sterilized for one hour under an ultraviolet lamp. The cells were detached, re-suspended in the growth medium and seeded on the NMP structures at a concentration of 25,000 cells/sample. Afterward, the seeded samples were incubated in standard conditions of temperature and humidity, for 24 h, to allow any possible attachment of the cells.

The in vitro studies were cut very short, i.e., during the 24 h of cell culture. The reason for this short time interval relies on the fact that the doubling time for the OLN cells is less than 24 h [65]. Moreover, the time necessary for the cells to attach to the surface is about 4 h. During the cell seeding procedure, the cells immediately drop onto the surface of the 3D structures due to the gravitational force; however, given the architecture of the structures, the cells are unable to form bonds, attach and spread onto the mushroom-like pillars. For this reason, they either fall from the surface or remain on the pillars and maintain a rounded shape, which is characteristic of non-attached cells.

#### 4.2.5. Immunocytochemistry

After 24 h of incubation, the cells were washed twice with phosphate-buffered saline solution (PBS), for 5 min, at room temperature and fixed with 3.7% paraformaldehyde (Chemical Company, Iasi, Romania) for 10 min. Afterward, the permeabilization was performed using a 0.1% Triton X-100 solution in PBS (Merck catalog number X100-100ML) for 10 min. The labeling of the cytoskeleton was conducted with 1:200 Texas Red™-X Phalloidin (Thermo Fisher catalog number T7471, Waltham, MA, USA), overnight, at 4 °C; the labeling for nuclei used 10 µg/mL Hoechst 33,342 (Thermo Fisher catalog number H1399), for 10 min, at room temperature.

#### 4.2.6. Image Acquisition and Processing

The images were acquired using an Olympus ×B51 microscope, equipped with a DSD2 module with a pE-4000 light source and Olympus UPLFLN objectives. The control software used was Andor IQ3 (Manchester, UK). 

Quantitative information regarding the cells’ number density and area was obtained using 617 ImageJ v1.36 (National Health Institute, Bethesda, MD, USA), using the function Area Fraction and Shape Descriptors. The results represent the means of 3 different experiments (in total 3 fields of view for each sample).

## Figures and Tables

**Figure 1 ijms-23-03247-f001:**
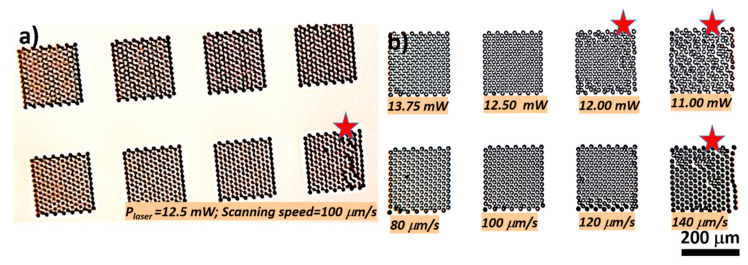
Optical images of arrays of nanostructured mushroom-like pillars (NMPs) arranged in hexagonal lattices, fabricated by LDW via TPP of IP-Dip photopolymer: (**a**) using the same geometry and writing parameters; (**b**) using different laser powers (upper panel) and scanning speeds (lower panel). Writing parameters (laser power, scanning speed) are indicated on each image. On both (**a**,**b**) the red stars mark the structures that lost their stability on the glass substrate.

**Figure 2 ijms-23-03247-f002:**
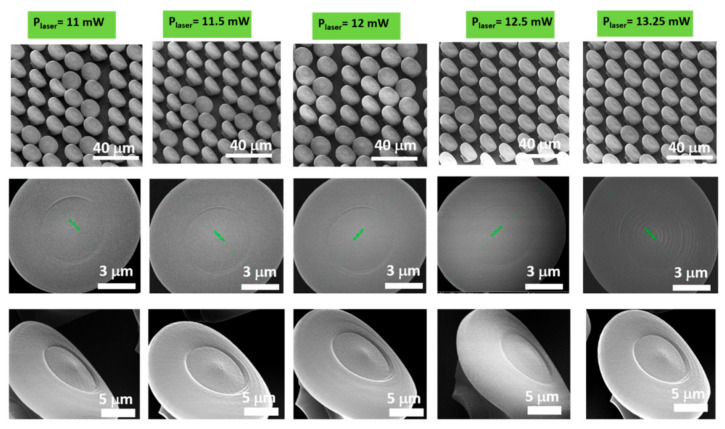
Scanning electron micrographs (SEM) of nanostructured mushroom-like pillars (NMPs), fabricated by LDW via TPP of IP-Dip photopolymer, imprinted using the same design and different laser powers, as indicated above each figure: (**Upper panel**) Tilted top views of NMPs arrays (tilted at 30 degrees); (**Middle panel**) Close top views of single NMP; (**Lower panel**) Tilted top views of single NMP (tilted at 30 degrees). For all samples, the scan speed was kept constant at 100 μm/s.

**Figure 3 ijms-23-03247-f003:**
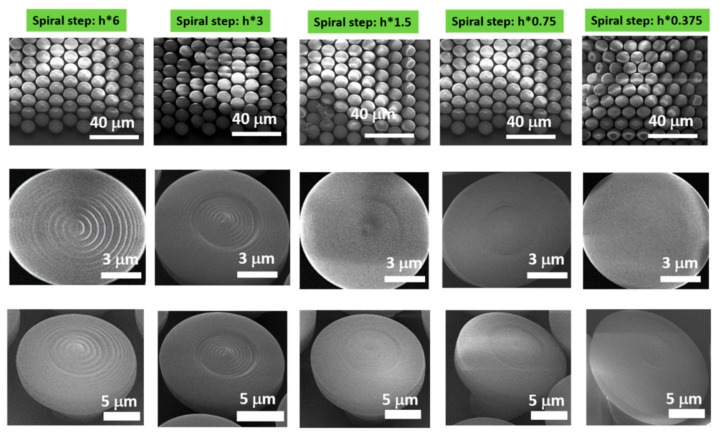
Scanning electron micrographs (SEM) of nanostructured mushroom-like pillars (NMPs), fabricated by LDW via TPP of IP-Dip photopolymer. The original spiral step (h) was multiplied with coefficients indicated above each figure: (**Upper panel**) Tilted top views of NMPs arrays (tilted at 30 degrees); (**Middle panel**) Close top views of single NMP; (**Lower panel**) Tilted top views of single NMP (tilted at 30 degrees). Writing parameters: 12.5 mW laser power and 100 μm/s scanning speed. The spiral step, h, represents the height difference of the laser focus after completing a 360-degree trip while writing the structure.

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
