# Peer review of "Laser Direct Writing of Dual-Scale 3D Structures for Cell Repelling at High Cellular Density"

_ijms, 2022, doi:10.3390/ijms23063247_

Round 1
Reviewer 1 Report
See attached.

Reviewer 2 Report
My comments are as follows:
- Section 2.1 is large; it is difficult to perceive information. It would be better if the authors break it up into several sections or reduce it.
- The authors should clearly separate the sections "Materials and Methods" and "Results and Discussion".
For example, the text in Lines 119-125, 231-240, 392-395 should be given in "Materials and Methods" or removed from the article.
- Lines 129-130: «Regarding the reproducibility, Figure 1a shows that most of the NMPs preserved their vertical position and spatial arrangement».
Figure 1a is of poor quality, so it is not possible to see the veracity of this sentence. Also, the writing parameters in Figure 1 are completely unreadable. Authors should improve the quality of the images and captions.
- Lines 179-180 «It can be seen that, for laser powers between 11 and 12 mW..»
& Line 184 «..Figure 2 middle panel, the “mushroom hats” were nearly the same for all NMPs»
& Lines 186-187 «..in Figure 2 lower panel, where it appears that, for laser powers between 11 and 12 mW, the "mushroom's hats" showed less surface undulation..»:
Based on the text for laser powers between 11 and 12 mW, the SEM turned out to be the same. It would be better to shorten Figure 2 and remove the duplicate images or bring them into Supplementary.
- Lines 254-262 or AFM results. It is better to give the numbers in the table and be sure to indicate the statistical deviation.
- Figure 4 needs to be redone or can be combined with Figure 2.
It is better to transfer the entire Lower panel to a graph of the dependence of periodicity/height on laser power.
Also check the dimension on the Lower panel: does it have micrometers in the rightmost picture?
The upper panel is completely unnecessary if the measurements were taken without considering the internal part of the structure.
- Figure 5: The data on the Middle panel should preferably be presented as a graph with the statistical deviation.
8. Figure 7а-с: Scale needs to be added, current photos are of poor quality.
9. Section 4.1: The authors should indicate clearly which parameters and within what limits they were changed during TPP, also indicate the dimensions of the three-dimensional model.
10. In vitro studies were cut very short (during 24h,). While this is some result, the work would have gained significantly, if the behavior of the cells had been observed a little longer.
11. Line 177 «..with different writing parameters (laser powers and scanning speeds)»
& Line 175 «the scan speed was kept constant at 100 mkm/s ».
It is necessary to clarify where the correct information is.
- Lines 66-68: «This further limited the control over the geometric characteristics (size, shape, orientation of nanopatterns), while also showing an increased sensitivity towards the processing parameters (incident wavelength, material type, laser power, laser pulse overlap, writing speeds).»
In this context, "Material type" does not refer to processing parameters.
